# Exosomes in Cerebral Ischemia-Reperfusion Injury: Current Perspectives and Future Challenges

**DOI:** 10.3390/brainsci12121657

**Published:** 2022-12-02

**Authors:** Chao Zhou, Fating Zhou, Yarong He, Yan Liu, Yu Cao

**Affiliations:** 1Laboratory of Emergency Medicine, Department of Emergency Medicine, West China Hospital, West China School of Medicine, Sichuan University, Chengdu 610041, China; 2Disaster Medical Center, Sichuan University, Chengdu 610041, China; 3Department of Emergency Medicine, Chengdu First People’s Hospital, Chengdu 610041, China; 4Department of Gastroenterology, The Fifth People’s Hospital of Chengdu, Chengdu 610041, China

**Keywords:** cerebral ischemia-reperfusion injury, exosomes, intercellular communication, stem cells

## Abstract

Cerebral ischemia impedes the functional or metabolic demands of the central nervous system (CNS), which subsequently leads to irreversible brain damage. While recanalization of blocked vessels recovers cerebral blood flow, it can also aggravate brain injury, termed as ischemia/reperfusion (I/R) injury. Exosomes, nanometric membrane vesicles, attracted wide attention as carriers of biological macromolecules. In the brain, exosomes can be secreted by almost all types of cells, and their contents can be altered during the pathological and clinical processes of cerebral I/R injury. Herein, we will review the current literature on the possible role of cargos derived from exosomes and exosomes-mediated intercellular communication in cerebral I/R injury. The PubMed and Web of Science databases were searched through January 2015. The studies published in English were identified using search terms including “exosomes”, “cerebral ischemia-reperfusion injury”, “brain ischemia-reperfusion injury”, and “stroke”. We will also focus on the potential therapeutic effects of stem cell-derived exosomes and underlying mechanisms in cerebral I/R injury. Meanwhile, with the advantages of low immunogenicity and cytotoxicity, high bioavailability, and the capacity to pass through the blood–brain barrier, exosomes also attract more attention as therapeutic modalities for the treatment of cerebral I/R injury.

## 1. Introduction

Cerebral I/R injury is the sudden onset of cerebral blood circulation disorders, including focal cerebral ischemia-reperfusion injury like stroke and global cerebral ischemia-reperfusion injury like cardiopulmonary resuscitation. Due to its high incidence, morbidity, and recurrence, it imposes a heavy economic burden on society and the health system [1]. There are two major types of stroke: cerebral ischemia and cerebral hemorrhage, and >87% of the stroke cases are of the ischemic type [2]. To date, the only therapeutic treatment for cerebral ischemia, approved by the FDA, is recombinant tissue plasminogen activator (rt-PA), which is used for recanalizing the blocked vessels [3]. However, its therapeutic efficacy is limited because of its short therapeutic window [4]. More importantly, there are also possible complications following revascularization therapy, leading to the failure of cerebral ischemia therapy. Among which, ischemia-reperfusion (I/R) injury is one of the most serious and unavoidable clinical issues [5]. Thus, an in-depth study of the mechanism of I/R injury is extremely important for disease management.

Intercellular communication between neurons, astrocytes, and vascular endothelial cells is essential for normal function of the brain, while this communication is interrupted after cerebral I/R injury [6]. Exosomes, nanovesicles with a size distribution between 30–150 nm, are released by the fusion of multi-vesicle endosomes (MVEs) with the plasma membrane, which mediates intercellular communication [7]. To date, exosomes are generated by almost all types of cells in CNS, including neural stem cells/progenitor cells, neurons, astrocytes, microglia, and oligodendrocytes [8,9]. Moreover, several studies support the idea that exosomes can act as a messenger of intercellular communication by being effective transporters of biological material, including nucleic acids, proteins, and metabolites, which play indispensable roles in brain homeostasis [10]. Over recent years, a growing number of studies have explored the fact that exosomes are stable in the circulation and could penetrate the blood–brain barrier (BBB). These properties confer the beneficial neuro-protective effects of exosome-based therapy following a cerebral I/R injury. [11,12]. In the present review, we mainly investigate the role of exosomes in cerebral repair after I/R injury and briefly discuss the therapeutic impact and potential applications of these cellular vesicles for cerebral I/R injury.

## 2. Characteristics of Exosomes

### 2.1. The Biogenesis of Exosomes

Exosomes, s extracellular membrane vesicles, originate from endosomes with a size of 30–150 nm in diameter, and their formation can be divided into three stages (Figure 1). The first step is the formation of “early” endosomes (EEs) via invagination of the cytoplasmic membrane. During the process, the proteins are transferred from the plasma membrane (PM) to the surface of the EEs. After that, the EEs sort and recycle the cargos via targeting to the endocytic recycling compartment, identified by Rab11, or delivery to late endosomes (positive for Rab7 and Rab9) [13]. For the second stage, intraluminal vesicles (ILVs) are formed by the inward budding of the endosomal membrane and accumulate in the lumen of the late endosome, where EEs mature into multivesicular bodies (MVBs) [14]. Finally, MVBs bind to lysosomes, leading to their degradation and recycling within the cell. In addition, MVBs also fuse with PM, which results in the release of ILVs to the cell surface as exosomes [7,10].

The molecular mechanisms involved in the biogenesis of exosomes can be divided into two types that are known as endosomal-sorting complex response for transport (ESCRT)-dependent or independent mechanisms. For the classical ESCRT pathway, the ESCRT-0 complex initiates the process by engaging the ubiquitinated cargos, subsequently binding to ESCRT to form the ESCRT-0/ESCRT-I complex. Then, the ESCRT-I worked together with the activated ESCRT-II complex to create and stabilize the vesicle neck. Finally, accompanied by the direct interaction with vacuolar protein sorting 25 (VPS25) from ESCRT-II or ALIX, the ESCRT-III complex is recruited at the ILV biogenesis site, which drives neck constriction [15]. Another ESCRT-dependent pathway involved in exosomal biogenesis is the syndecan-syntenin-ALIX pathway [16]. However, the exosomes are still formed when the ESCRT is knocked down, suggesting that the ESCRT-independent system is also involved in exosome biogenesis [17,18]. With regard to the two ESCRT-independent systems, tetraspanins (CD9, CD63, CD81, and CD82) and lipids (ceramide, cholesterol, and phosphatidic acid) are important for exosome biogenesis [10,19,20].

### 2.2. Contents of Exosomes

Exosome posse lipid bilayer membrane structures and cargo biological materials (Figure 2). Specifically, lipids rich in exosomes, such as cholesterol, ceramide, and sphingomyelin, are essential for the membrane structure of exosomes as well as exosome secretion [21]. In addition, exosomes contain a variety of membrane-associated, high-order oligomeric protein complexes, including tetraspanins, protein membrane transport fusion proteins, transmembrane proteins, and heat shock proteins [22,23]. Moreover, exosomes also contain nucleic acids, such as DNA, mitochondrial DNA (mtDNA), mRNA, microRNAs, and other non-coding RNAs, and the transfer of nucleic acids from donor cells can regulate the biochemical reactions of recipient cells [24,25]. The intercellular communication mediated by the horizontal transfer of genetic information plays a crucial role during the development of disease.

### 2.3. Exosomes Uptake in Recipient Cells during Intercellular Communication

Exosome-mediated intercellular communication occurs via direct interaction or indirect interaction with recipient cells (Figure 3). Exosomes bind to target cells via ligand–receptor interactions, including proteins (glycoproteins, integrins, and tetraspanins), sugar (heparan sulfate proteoglycans), and lipids, facilitating the subsequent endocytosis. For example, when the transmembrane protein was degraded after the treatment with trypsin or proteinase K, exosome uptake by target recipient cells was significantly inhibited, indicating the crucial role of transmembrane proteins in exosome-recipient cell communication. [26,27]. Once exosomes dock at the surface of the recipient cells, cargos in exosomes are transferred into recipient cells by endocytosis routes (clathrin-mediated endocytosis [28], caveolin-mediated endocytosis [29], phagocytosis [30], lipid raft dependent internalization [31], and macropinocytosis [32], as well as direct fusion with the plasma membrane [33]. Moreover, exosomes can mediate intercellular communication by indirect interaction. As some ligands cannot be directly recognized by the membrane receptors, this indirect interaction needs a further cleavage of ligands on the exosome surface in order to bind to the receptors on the recipient cell surface. For instance, transmembrane proteins on the surface of exosomes can be cleaved by proteases to produce soluble forms of protein and subsequently bind to the receptors on the recipient cell surface [34].

## 3. Exosomes Profile Changes after Is Chemia-Reperfusion in Brain

In the brain, exosomes can be released by almost all cell types, including neurons, astrocytes, oligodendrocytes, endothelial cells, as well as microglia [35]. Recently, numerous studies have suggested that exosome-mediated cell communications are closely associated with the physiological and pathological processes in cerebral I/R injury (Figure 4) [23,36].

### 3.1. Neurons-Derived Exosomes

Neurons are a basic and functional unit of the CNS; neuron-derived, exosome-mediated, trans-synaptic communication plays an important role in neuronal functions, especially synaptic activity and neural circuit development [37]. However, under physiological and pathological conditions, a dynamic change of the contents of neuronal exosomes is reported. Chiang et al. have found that 45 microRNAs were significantly different in neuronal exosomes between normoxic and ischemic reperfusion stimuli. Moreover, as a consequence, neurons-secreted exosomes could impair neuronal cell viability and reduce neurite outgrowth, compared with exosomes from normoxic conditions [38]. In addition, Men et al. have confirmed that exosomes secreted by neurons could be internalized into astrocytes to regulate astrocyte functions under physiological conditions [39]. In the I/R injury model, miR-181c-3, loaded in neurons and cortical neurons-derived exosomes, internalized into astrocytes, thereby inhibiting neuro-inflammation in astrocytes via directly downregulating CXCL1 [40]. Moreover, neuron-derived exosomes enriched with miR-21-5p could be phagocytosed by microglia, which promoted microglia M1 polarization, thus leading to the aggravation of neuroinflammation [41].

### 3.2. Astrocytes-Derived Exosomes

Astrocytes play a pivotal role in the formation of the myelin sheath, structural integrity of synaptic, synaptic transmission, and neuronal function by actively clearing neurotransmitters in synapses, maintaining the structure of the blood–brain barrier, and releasing trophic factors to neurons [42]. Following cerebral I/R injury, astrocytes underwent a significant transformation called “reactive astrocytosis” [43]. In the CNS system, astrocytes also communicate with neurons and other glial cells via secreting exosomes. For example, astrocyte-derived exosomes promoted the proliferation of neurons and alleviated neuronal injury via transferring miR-34c [44]. Similarly, miR-361 in astrocyte-derived exosomes elevated cell activity and suppressed neuronal apoptosis in vitro and alleviated neurological deficits in rats with cerebral I/R injury [45]. In addition, activated astrocytes could modulate microglial polarization via exosome-mediated cargo transfer, which plays a vital role in neuroinflammation. Specifically, astrocyte-derived exosomes enriched with miR-873a-5p could be taken up by microglia and elicit microglia polarization into a M2 phenotype. As a result, the brain defect area and neurological deficits in the brain injury model were alleviated via inhibiting neuroinflammation [46].

### 3.3. Oligodendrocytes-Derived Exosomes

In the brain, oligodendrocytes are the myelinating cells in the CNS, and loss of oligodendrocytes induces demyelination, which leads to impaired neurological function in the event of a stroke [47]. A previous study has confirmed that secretion of oligodendrocyte-derived exosomes was initiated by glutamate, which was released by electrically active neurons [48]. Moreover, researchers found that upon stimulation neurons could uptake oligodendrocyte-secreted exosomes, thus conferring the enhancement of neuronal stress tolerance via promoting neuronal survival in the cerebral ischemic model [49,50]. In addition, oligodendrocyte-derived exosomes could also be specifically and efficiently taken up by microglia via a micropinocytosis mechanism and then transported to the lysosomes for degradation, suggesting that microglia are essential for the degradation of oligodendroglial membranes via the macropinocytotic clearance process [51]. Based on this observation, Peferoen et al. have described that the cross-talk mediated by exosomes between oligodendrocytes and microglia in the CNS system controlled the neuro-inflammation in neurodegenerative disorders [52].

### 3.4. Endothelial Cells-Derived Exosomes

In the brain, intercellular communication in neurovascular units (NVUs) was essential for CNS homeostasis and function. In NVC, endothelial cells maintain the dynamic balance and form the BBB, which precisely controls the transport of macromolecules and nutrients [53,54]. Available data indicated that exosomes from endothelial cells could alter the environment in the NVU after a stroke. For example, Xiao et al. have demonstrated that exosomes released from endothelial cells protected neurons from I/R injury in vitro via suppressing the I/R-induced cell cycle arrest and apoptosis, as well as improving cell proliferation, migration, and invasion [55]. Moreover, in the acute middle cerebral artery occlusion (MCAO) model, exosomes released from endothelial cells could accelerate neural progenitor cell (NPC) proliferation and migration and subsequently reconstruct the NVU [56]. On the contrary, following I/R injury, in response to the release of danger-associated molecular patterns (DAMPs) from necrotic neurons, the function of endothelial cells expressing pattern recognition receptors (PRRs) was impaired, leading to increased vascular permeability [57,58]. After that, endothelial cell-derived exosomes crossed the BBB, resulting in a detrimental inflammatory cascade and neuronal destruction via recruiting microglia to the sites of tissue injury [59].

### 3.5. Microglia-Derived Exosomes

Microglia, gatekeepers of the immune system in the CNS, surveil the cerebral environment constantly, which plays a key role in maintaining normal brain function and homeostasis [60]. Microglia adopt distinct and different phenotypes depending on the extracellular environment and can be roughly distinguished as two subtypes: classically activated microglia (CAM, M1-like microglia) and alternatively activated microglia (AAM, M2-like microglia). Generally, M1 microglia secrete pro-inflammatory factors and exacerbate brain tissue damage, while M2 phenotype microglia secrete anti-inflammatory cytokines and facilitate the recovery following cerebral I/R injury [61].

Recent studies have confirmed that microglia-derived exosomes are deeply involved in intercellular communication in the CNS system after cerebral I/R injury [62]. Microglia-derived exosomes can be divided into two types, including M1 microglia-derived exosomes (defined as detrimental exosomes) and M2 microglia-derived exosomes (defined as beneficial exosomes). In the event of a stroke, miR-383-3p-enriched exosomes from M1 microglia promoted the necroptosis of neurons via negatively regulating the expression of activating transcription factor 4 (ATF4) [63]. Moreover, miR-424-5p in M1 microglia-derived exosomes could be shuttled to cerebral endothelial cells after ischemic stimuli, which induced cell injury of viability and permeability via suppression of fibroblast growth factor 2 (FGF2)/signal transducers, and activators of the transcription 3 (STAT3) pathway in endothelial cells [64]. Contrary to M1 microglia, exosomes from M2 microglia exhibit neuroprotective effects after I/R injury via promoting neuronal survival [65,66] and tube formation of endothelial cells [67].

## 4. Effects of Exosomes in Cerebral I/R Injury

Exosomes could transport cell type-specific cargo extracellularly by long-range communication in the CNS. Beside the intercellular communication between different cells during the process of cerebral I/R injury, the altered exosomal number and contents after I/R injury may mediate beneficial effects, including neurogenesis, angiogenesis, and immune regulation, which synergistically accelerate the NVU reconstruction and neurological recovery (Figure 5).

### 4.1. Exosomes Effects on Neurogenesis

In cerebral I/R injury models, axons are reduced. Therefore, neurogenesis, or the birth of new neurons, which is known to be induced in the infarct and surrounding areas, is beneficial to spontaneous neurologic improvement after I/R injury [68]. Exosomes secreted by different cells in the NVU regulate both regeneration and repair of central nervous system circuits. Endothelial progenitor cells (EPCs), with the potential for differentiation into mature endothelial cells, are essential to repair endothelial damage. Along with the angiogenesis after cerebral I/R injury, EPC-derived exosomes tend to be internalized by neurons and promote neurogenesis via suppressing ischemia-injured apoptosis, which attributes to the enrichment of miR-126. Moreover, the author also demonstrated that exosomes from miRNA-126-modified EPCs were more effective than those from EPCs in decreasing ischemic injury in diabetic stroke mice via elevating the expression of vascular endothelial growth factor receptor 2 (VEGFR2) [69]. As one special type of stem cell, neural stem cells (NSCs) only exist in the nervous system, and they can differentiate into different types of nerve cells in the CNS, such as neurons, astrocytes, and oligodendrocytes, with the ability to compensate for insufficient endogenous nerve cells [70]. Zhang et al. have highlighted that exosomes derived from inflammatory factor IFN-γ-stimulated neural stem cells (NSCs) could promote neurogenesis in an ischemic stroke rat model due to the enrichment of miR-206, miR-133a-3p, and miR-3656 [71]. In addition, the exosomes derived from microglia were elevated after I/R injury, which are mainly associated with primary neurons and neurite processes [69]. Moreover, another study has found that nervous growth/differentiation factor (nGDF) enriched in microglia-derived exosomes exhibited significant neurotrophic activities to promote nerve regeneration and recovery [72]. Similarly, Song et al. have detected that after exosomal transfer of miR-124 from M2 microglia to neurons, neuronal survival and neurogenesis were promoted so as to protect the mouse brain from I/R injury [65]. Additionally, Tassew et al. have found that exosomes released from fibroblasts could activate the Wnt family member 10b (Wnt10b)-mammalian target of rapamycin (mTOR) pathway, which subsequently restored the intrinsic neuronal regeneration [73]. In summary, it is worth investigating the neurogenesis effects of exosomes, which are thus considered as an intervention for ischemic stroke.

### 4.2. Exosomes Effects on Angiogenesis

Therapeutic angiogenesis is vital in improving vascular functions and maintaining BBB homeostasis following I/R injury in the brain. Wang et al. have revealed that EPC-derived exosomes improved angiogenesis in diabetic ischemic stroke mice, and enhanced therapeutic efficacy was obtained by miR-126 enrichment [69]. Moreover, after the delivery of miR-132 from neuron-derived exosomes to endothelial cells, an elevation of vascular endothelial cadherin (VE-cadherin) expression was observed, which could improve the cerebral vascular integrity and functions after cerebral ischemic injury [74]. Additionally, delta-like ligand 4 (Dll4) proteins in exosomes obtained from human microvasculars promoted angiogenesis via activation of the Dll4–Notch signaling pathway, which is crucial for vascular development and angiogenesis [75]. Furthermore, with the treatment of M2 microglia, the brain injury induced by I/R was ameliorated by promoting angiogenesis, and the beneficial effects relied on the release of exosomes enriched with miRNA-26a [67]. Accordingly, various cell-derived exosomes play a beneficial effect in restoring the angiogenesis, thus improving cerebral I/R injury.

### 4.3. Exosomes Effects on Immune Regulation

As one of the critical pathogenic mechanisms in cerebral I/R, inflammation causes cascade injury to the brain. Microglia are believed to be the most critical immune defense, and the microglial polarization into different phenotypes is closely related to the inflammation in the cerebral I/R injury model [76]. It has been observed that exosomes possess miRNA could directly modulate the levels of toll-like receptors (TLRs) or nuclear factor-κB (NF-κB). They can inhibit the M1 microglia polarization and subsequently reduce the level of inflammatory cytokines, which in turn alleviates the cerebral I/R injury [77,78]. Meanwhile, exosome-mediated M2 polarization could elevate the secretion of anti-inflammatory cytokines, including interleukin-4 (IL-4), interleukin-10 (IL-10), interleukin-5 (IL-5), transforming growth factor-β(TGF-β), and neurotrophic factors, which are beneficial to the recovery of brain function and the improvement of the prognosis of strokes [79,80,81]. Among them, IL-4 not only elevates microglial phagocytosis via activation of peroxisome proliferator-activated receptor γ (PPARγ) to enable the effective cleanup of apoptotic neurons but also produces neurotrophic factors, which are essential for brain repair [82]. Additionally, compared to mild stroke patients, the IL-5 level is decreased in severe stroke patients with poor outcomes and may be used as a predictor of edema and infarct volume [2]. Moreover, IL-5 could inhibit proinflammatory gene expression, including inducible nitric oxide synthase (iNOS) in microglia, and subsequently suppress the neuroinflammation in the cerebral I/R model [83]. Apart from microglia, miR-34c released from astrocyte-released exosomes exerted a protective role against neurological deficits via inhibiting TLR7 and NF-κB/(mitogen-activated protein kinase) MAPK pathways [55]. Moreover, endothelial-derived miR-199a-5p could protect neural cells against ER stress-caused inflammation by targeting binding immunoglobulin protein (BIP) [84]. Therefore, inhibition of inflammation is pivotal to protecting the brain against I/R injury, and it may be beneficial to target specific exosomes related to it.

## 5. Exosomes-Based Therapy and Application in Cerebral I/R Injury

As a crucial paracrine way of cell therapy, exosomes have attracted wide attention to treat cerebral I/R injury due to their unique characteristics, including lower immunogenicity, minimal oncogenicity, reduced chance of vascular blockage, and capacity to cross the BBB, and their key role in intercellular communication [36,85]. Moreover, only a few cells can secrete abundant exosomes, and they can be stored stably [23]. The therapeutic potential of exosomes is mainly dependent on transferring their cargos to receipt cells in the CNS, especially microRNAs. Eventually, engineered exosomes carried with modified microRNAs tend to activate the regeneration of the CNS and the recovery of neurological function more efficiently [22]. Moreover, 98% of small molecular drugs do not cross the BBB and effectively arrive at injured sites in the brain, resulting in inefficient targeting, low release power, and failure to reach therapeutic concentrations in the brain [86]. Exosomes are used for the delivery of drugs (such as curcumin and enkephalin), which offer significant protective effects in cerebral I/R injury [87,88]. Furthermore, surface modification of exosomes additionally improves exosomal functions, thus further enhancing specific cell targeting [89,90], in vivo imaging [91], and tracking [92].

Stem cell-based therapies have been confirmed to promote recovery in cerebral I/R injuries. Numerous studies suggest that stem cell-related therapeutic effects are mainly mediated via the paracrine mechanisms, among which exosomes are intensively explored. Moreover, compared with stem cells, stem cell-derived exosomes may be a potential therapeutic option due to their unique characterizations, including higher biocompatibility, stable biological properties, and low immunogenicity. Recently, therapeutic benefits of exosomes released by stem cells, including mesenchymal stem cells (MSCs), NSCs, and induced pluripotent stem cells (iPSCs), have been reported in the cerebral I/R injury model (Table 1).

### 5.1. MSCs-Derived Exosomes

MSCs are pluripotent stem cells with the ability of multilineage differentiation, and MSC-based therapy has proven potential as an effective therapeutic option for ischemic strokes [93]. Thorsten et al. have demonstrated that MSC-derived exosomes promote neurological recovery after strokes, which is similar to the effects of MSCs [94]. Based on this observation, an increasing number of studies have been performed to investigate the role of exosomes obtained from MSCs in cerebral I/R injuries. The intercellular exosomes-mediated communication between MSCs and brain parenchymal cells was observed to promote neurite growth via transferring miR-133b to neurons and astrocytes [95,96]. Moreover, MSC-derived exosomes transferred miR-17-92 to recipient cells, which led to improvements in neurogenesis, neuroplasticity, and oligodendrogenesis in MCAO models via targeting the phosphatase and tensin homolog (PTEN)/Akt pathway [97]. In addition, MSC-derived exosomes, with robust angiogenic paracrine factors, promoted endogenous angiogenesis in the brain via inhibition of the NF-κB pathway [98]. Furthermore, protective effects in cerebral I/R injury were obtained by MSCs-exosomal pigment epithelium-derived factor (PEDF), which was dependent on the promotion of autophagy [99]. At the same time, apart from the neurogenesis and angiogenesis, MSCs-exosomes could elicit M1 microglia into the M2 phenotype and inhibit the inflammation mediated by the NLRP3 inflammasome, thus alleviating the cerebral I/R injury [79,100], suggesting the immune modulation of MSCs-exosomes. Similarly, miR-223-3p-enriched MSC-derived exosomes suppressed microglial M1 polarization induced by the cysteinyl leukotriene receptor 2 (CysLT2R)-ERK1/2 pathway, in mice with cerebral I/R injuries [101]. In addition to the regulation of microglia polarization, other mechanisms of anti-inflammatory action have also been discovered. For example, MSC-derived exosomal miR-146a-5p inhibited neuro-inflammation and neurological deficits by inhibition of the interleukin 1 receptor associated kinase 1 (IRAK1)/TNF receptor associated factor 6 (TRAF6) pathway in ischemic stroke [102]. Moreover, miR-138-5p/LCN2, miR-221-3p/activating transcription factor 3 (ATF3), and the miR-26b-5p/cholesterol 25-hydroxylase (CH25H) pathway have also been detected, which contribute to MSCs-exosome conferred inhibition of inflammation in response to the cerebral I/R injury [77,103,104]. More interestingly, due to the enrichment of some certain functional proteins in exosomes, exosomes derived from MSCs in I/R brain extract, oxygen glucose deprivation, or hypoxic pretreatment exerted better neuroprotective effects [81,105,106].

Recently, coronavirus disease 2019 (COVID-19), a global epidemic pneumonia, predisposed hospitalized patients to cerebral I/R injury, which was closely correlated with a hypercoagulable state induced by systemic inflammation. Researchers have discovered that cells infected with severe acute respiratory syndrome coronavirus 2 (SARS-CoV-2) virus could yield exosomes enriched with viral proteins [107,108]. Moreover, spike, a SARS-CoV-2 product, has the ability to modify the cargo in host cell-derived exosomes. As a result, along with the cargo transfer to distant uninfected organs, it caused a cascading inflammatory response within the CNS [108,109]. Due to the immunoregulatory characteristics of MSCs, MSC-derived exosomes can also be used, at least as supportive treatment, in COVID-19 patients to protect against the neuroinflammation and subsequent cerebral I/R injury caused by a cytokine storm or direct CNS infection [110].

### 5.2. NSCs-Derived Exosomes

NSCs, self-renewing and endogenous multipotent cells in the brain, have the capacity to differentiate into multiple neural cell types, including neurons, astrocytes, and oligodendrocytes. After brain injury, quiescent NSCs are activated and initiate the self-repair process. In preclinical and clinical research, NSC-based therapy has shown potential for the regenerative treatment of cerebral I/R injury [111,112]. For example, NSC transplantation in the CNS regions could restore brain homeostasis after I/R injury via preservation of the BBB, suppressing the inflammation, and improvement of the neurogenesis and angiogenesis [113]. Moreover, numerous animal studies have confirmed that bystander effects include delivery of NSC-derived therapeutic gene products to regulate the extracellular microenvironment and promote neuronal circuit plasticity [114]. NSC-derived exosomes act as an important vehicle to deliver therapeutic agents and show the neuroprotection effects of cerebral I/R injuries [71]. It has been confirmed that multiple tail vein injections of NSC-conditioned medium effectively suppressed neuronal apoptosis while maintaining mitochondrial homeostasis so as to ameliorate cerebral I/R injuries in rats [115]. Pluchino et al. have revealed that interferon-γ (IFN-γ) and tumor necrosis factor-α (TNF-α) regulated the phenotype of stem cells, soluble factors secreted from cells, and ultimately altered the functions of stem cells [116]. Interestingly, without effects on NSC proliferation, IFN-γ elevated the superoxide dismutase 2 (SOD2) level in NSC culturing, which improved the therapeutic effects of NSCs in the ischemic stroke model [117]. Beside the altered characteristics of stem cells, including the phenotype and secreted proteins, IFN-γ preconditioning further exerted therapeutic effects via carrying specific exosomal miRNA cargos (miR-206, miR-133a-3p and miR-3656), although the secretion and characteristics of exosomes derived from NSCs were not affected [71].

### 5.3. IPSCs-Derived Exosomes

iPSCs were first discovered by Takahashi and Yamanaka in 2006 from adult somatic cells via integrating four factors (Sox2, Oct3/4, Klf4, and cMyc) [118]. Till date, with the great capacity of self-renewal and differentiation into all somatic cell types, iPSC-based cell therapy has been widely investigated, and their potential in the treatment of neurodegenerative diseases has been studied [119,120]. Rajasingh et al. have compared human iPSC-derived MSCs (iMSCs) and MSCs and found that iMSCs still maintained MSC characteristics without any chromosomal abnormalities even at later passages, while MSCs started to lose their characteristics at that time, indicating that iMSCs might be an alternative cell type to MSCs for the treatment of various diseases [121]. According to the success of MSC-derived exosome therapy against ischemic strokes, Xia et al. have revealed that in ischemic strokes, iMSC-derived exosomes could improve angiogenesis, potentially via STAT3 activation-mediated autophagy inhibition [122]. In addition, accumulating researchers have found that NPC transplantation promoted functional recovery in cerebral I/R injury. Whereas, the NPCs have mostly been harvested from embryonic stem cells or fetal tissue, which raises ethical issues. Recent studies have revealed that, compared with other stem cells, human iPSC-derived neural progenitor cells (iNPCs) are more similar to cortical neurons in morphology and immunohistochemistry [123]. Moreover, iNPC-exosome treatment exhibited a neuroprotective effect in a porcine model of ischemic stroke via promoting neurite outgrowth [124].

Acronyms: MSCs, mesenchymal stem cells; NSCs, neural stem cells; iPSCs, induced pluripotent stem cells; CH25H, cholesterol 25-hydroxylase; PTEN, phosphatase and tensin homolog; NF-κB, nuclear factor-κB; IRF5, IFN regulatory factor 5; CysLT2R, cysteinyl leukotriene receptor 2; IRAK1, interleukin 1 receptor associated kinase 1; TRAF6, TNF receptor associated factor 6; ATF3, activating transcription factor 3; IL-10, Interleukin 10; TGF-β1, transforming growth factor β1; STAT3, signal transducer and activator of transcription 3.

**Table 1 brainsci-12-01657-t001:** Studies of stem cell-derived exosomes in cerebral I/R injury.

Stem Cells	Contents	Mechanism	Function	Refs
MSCs	miR-26b-5p	CH25H	Microglial M1 polarization	[77]
	miR-133b	N/A	Neurite outgrown	[95,96]
	miR-17-92	PTEN/Akt	Neural plasticity	[97]
	N/A	NF-kB	Angiogenesis	[98]
	PEDF	Autophagy	neuronal apoptosis	[99]
	miR-22-3p	IRF5	Microglial M1 polarization	[100]
	miR-223-3p	CysLT2R	Microglial M2 polarization	[101]
	miR-146a-5p	IRAK1/TRAF6	Neuro-inflammation	[102]
	miR-138-5p	lipocalin 2	Proliferation of astrocytes Inflammation	[103]
	miR-221	ATF3	Inflammation Neuronal apoptosis	[104]
NSCs	N/A	Bcl-2	Apoptosis Mitochondrial ultrastructure	[115]
	N/A	IL-10, TGF-β1	Inflammation	[117]
iPSCs	N/A	STAT3	Angiogenesis	[122]
	N/A	PTEN/Akt	neurite outgrowth	[124]

### 5.4. The Limitations of Exosomes-Based Therapy in Cerebral I/R Injury

To date, a clinical trial related to the use of exosomes as a stroke treatment has been carried out, in which miR-124-enriched MSCs-exosomes were administered to patients with acute ischemic strokes (ClinicalTrials.gov: NCT03384433). However, there are still some problems that need to be conclusively solved in this field. Firstly, exosome cargos depend on donor cells, culture conditions, and the methods of exosome separation. Thus, efficient separation technology, a quality control standard for clinical-grade exosomes, and the characterization of exosome cargos endowing therapeutic potential is warranted. Secondly, the dosage, frequency, and administration routes of exosome delivery still do not have concise agreement, so further investigation is necessary. Thirdly, bioengineered ligands or modifications of exosomes to extend the half-life of exosomes and improve their targeting ability in vivo need to be performed. Lastly, at present, the potential adverse effects of exosomes in patients are unknown. Therefore, long-term and large-scale clinical trials are needed to allow the translation of exosome therapy into clinical practice.

## 6. Conclusions and Prospect

Cerebral I/R injury is one of the main causes of morbidity and mortality in the world. Exosomes have attracted considerable attention due to their unique biological properties. As a communication substance and transport carrier, exosomes play important roles in cerebral I/R injury, such as neurogenesis, angiogenesis, and immune regulation, which coordinately promote NVC reconstruction and neurological recovery. Recent reports have focused on the application of exosomes as a potential drug delivery approach in cerebral I/R injury. In this review, we highlight the biological roles of exosomes and exosome-mediated intercellular communication in cerebral I/R injury. In spite of the critical role of exosomes during the pathologic processes of cerebral I/R injury, exosomes have been reported as a biomarker for diagnosis in brain-related diseases, which was limited in this review. To date, the exosome study remains in its initial stages, particularly for cerebral I/R injury, and not enough information is available to translate exosome treatment into clinical practice. In summary, exosome-based therapy has great potential when it comes to alleviating brain damage following I/R injury, which is worthy of further investigation.

## Figures and Tables

**Figure 1 brainsci-12-01657-f001:**
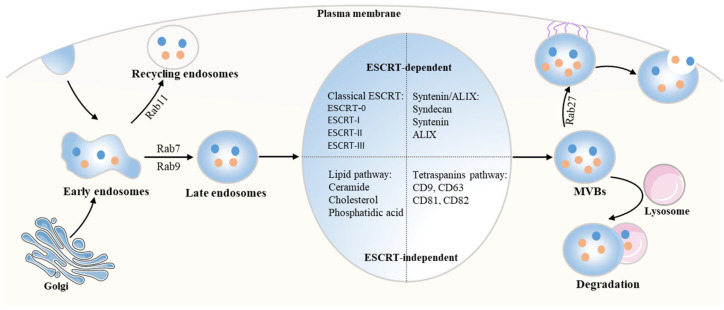
**Schematic representation of exosome biogenesis.** The early endosomes (EEs) are formed through the invagination of the plasma membrane, in which the transmembrane proteins are sorted to late endosomes by Rab7 and Rab9. Then, the EEs further generate intraluminal vesicles (ILVs), which lead to the formation of multivesicular bodies (MVBs). There are two types of molecular mechanisms involved in the process: Endosomal Sorting Complex Response for Transport (ESCRT)—dependent or independent pathways. After that, with the help of Rab27, the MVBs dock and fuse with the plasma membrane to release it as exosomes. Additionally, the MVBs can also fuse with lysosomes for degradation and recycling. Acronyms: Rab7, Ras-related protein 7; Rab9, Ras-related protein 9; ILVs, intraluminal vesicles; MVBs, multivesicular bodies; ESCRT, Endosomal Sorting Complex Response for Transport; Rab27, Ras-related protein 27.

**Figure 2 brainsci-12-01657-f002:**
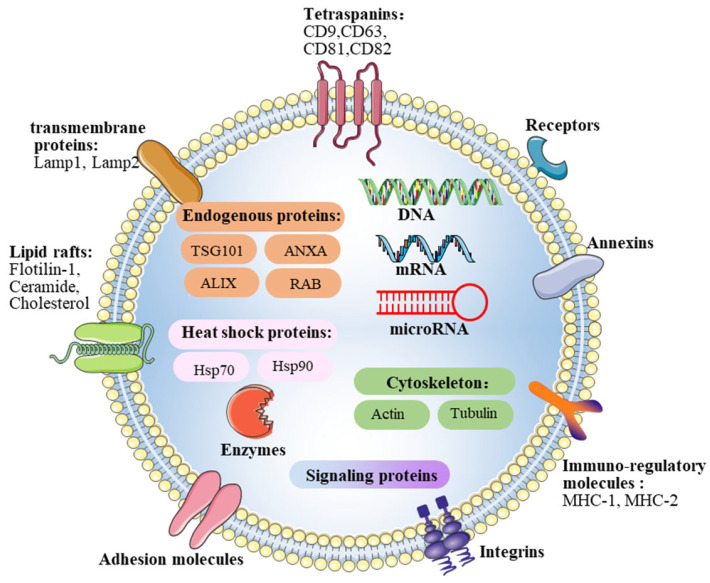
**Contents of exosomes.** As lipid bilayer membrane vesicles, exosomes are rich in cholesterol, phospholipids, and ceramides. Exosomes contain conserved proteins, such as tetraspanins, integrins, immune-regulatory molecules, protein membrane transport fusion proteins, as well as transmembrane proteins (Lamp1, Lamp2). In addition, exosomes contain many biologically active molecules, including enzymes, DNA, mRNA, and microRNA, etc.

**Figure 3 brainsci-12-01657-f003:**
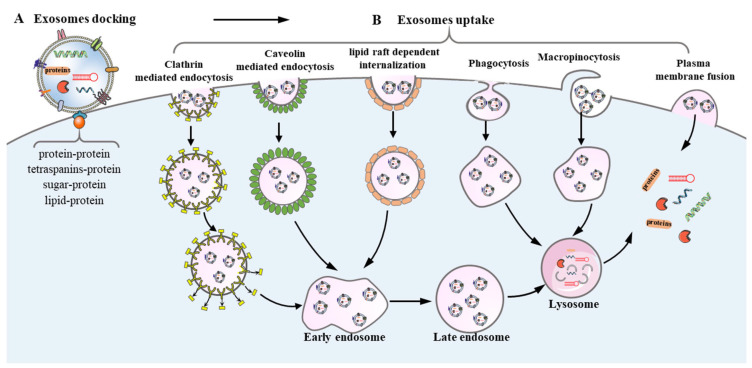
**Mechanisms of Exosome Uptake.** (**A**) When exosomes reach the recipient cells, they can dock at the plasma membrane of the recipient cells through the interaction between exosomal surface proteins and receptors on recipient cells. (**B**) After that, the exosomes may be taken up by recipient cells via the endocytosis route (clathrin mediated endocytosis and caveolin-mediated endocytosis), phagocytosis micropinocytosis, receptor-mediated endocytosis, as well as by direct fusion with the plasma membrane, which causes the release of the contents into the cytoplasm.

**Figure 4 brainsci-12-01657-f004:**
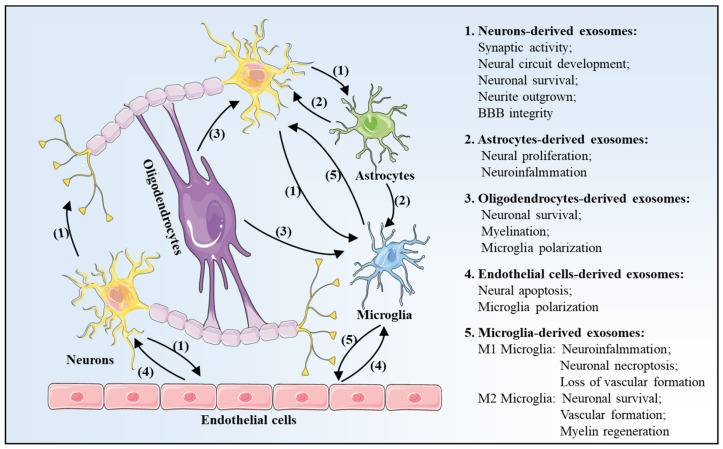
Schematic diagram of intercellular communication mediated by exosomes in the central nervous system (CNS) after ischemia-reperfusion (I/R) injury. Exosomes can be released by almost all cells in the CNS, including neurons, endothelial cells, astrocytes, oligodendrocytes, and microglia, and they are involved in the process of cell-to-cell communication. Acronyms: I/R, ischemia-reperfusion; CNS, central nervous system. CNS. Arrows indicate the direction of transfer.

**Figure 5 brainsci-12-01657-f005:**
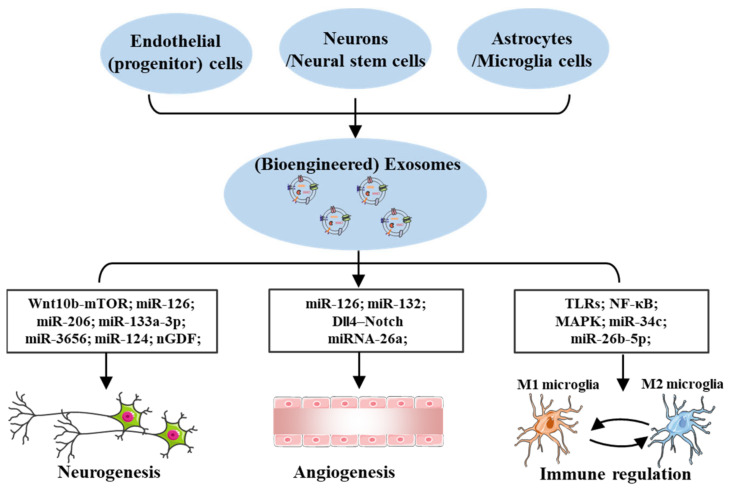
**Exosomes-mediated effects in cerebral ischemia-reperfusion (I/R) injury.** Exosomes derived from various cells in the central nervous system (CNS) play a pivotal role in neurogenesis and angiogenesis following I/R injury. In addition, the inhibition of M1 microglia polarization or the reprogramming of microglia from M1 to M2, mediated by exosomes, is important for the suppression of inflammation in cerebral I/R injury. Acronyms: Wnt10b, Wnt family member 10b; mTOR, mechanistic target of rapamycin; nGDF, nervous growth/differentiation factor; TLRs, toll-like receptors; NF-κB, nuclear factor-κB; MAPK, mitogen-activated protein kinase.

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
