# Peer review of "Exosomes in Cerebral Ischemia-Reperfusion Injury: Current Perspectives and Future Challenges"

_brainsci, 2022, doi:10.3390/brainsci12121657_

Round 1
Reviewer 1 Report
Review for the manuscript “Exosomes in cerebral ischemia-reperfusion injury: Current Perspectives and Future Challenges”
Dear authors, I thank you for the opportunity to review this interesting manuscript. In my opinion, it requires some corrections before it can be accepted for publication.
ABSTRACT
In lines 19-27 we can find: “Herein, we will provide an overview of the current literature on the possible role of cargo derived from exosomes, and exosomes-mediated intercellular communication in cerebral I/R injury. Meanwhile, with the characteristics of low immunogenicity, high transmission efficiency, and the ability to cross the blood–brain barrier (BBB), exosomes also attract more attention as therapeutic modalities for the treatment of cerebral I/R injury. In this review, we also focus on the potential therapeutic effects of stem cells-derived exosomes and underlying mechanisms in cerebral I/R injury”
I have three comments on the above:
1) The authors write “blood–brain barrier (BBB)”. The acronym is unnecessary since it is cited only one time.
2) I suggest re-write as follows:
“Herein, we will provide an overview of the current literature on the possible role of cargo derived from exosomes, and exosomes-mediated intercellular communication in cerebral I/R injury. We also will focus on the potential therapeutic effects of stem cells-derived exosomes and underlying mechanisms in cerebral I/R injury. Meanwhile, with the characteristics of low immunogenicity, high transmission efficiency, and the ability to cross the blood–brain barrier, exosomes also attract more attention as therapeutic modalities for the treatment of cerebral I/R injury.”
3) If it is possible (see the allowed number of words to the abstract), include more results.
INTRODUCTION
In line 29-30 we can find “Cerebral ischemia-reperfusion injury is a sudden onset of cerebral blood circulation…”. Since in the abstract the authors used I/R, I suggest changing the sentence for “Cerebral ischemia-reperfusion (I/R) injury is a sudden onset of cerebral blood circulation…”. In line 42 the authors use “ischemia-reperfusion (I/R) in…” but here is the second time that the term was used.
In line 56-57 we see: “In the present review, we 56 highlight the role of exosomes in cerebral repairing processes and discuss the therapeutic 57 impact and potential applications of exosomes for cerebral I/R injury.” I suggest changing for “In the present review, we investigate the role of exosomes in cerebral repairing processes and discuss the therapeutic impact and potential applications of these cellular vesicles for cerebral I/R injury”
I congratulate the authors for the very nice Figures.
There are acronyms in Figure 1 that are not defined in the Figure legend. Please check the figures regarding this comment.
Please, do the same for Table 1. Include legend (for example, what is the meaning of NFKB and TGF-beta?)
In line 194-195 we find: “For example, Bing Xiao et al have demonstrated that exosomes released from endothelial cells protected neurons from I/R injury in vitro [55]”. Please, explain how this protection happens.
In lines 242-244 we can read that “Along with the angiogenesis after cerebral I/R injury, EPCs-derived exosomes tend to be internalized by neurons and promote neurogenesis via suppressing ischemia-injured apoptosis”. Please, include more information on this statement.
The definition of transforming growth factor beta (TGF-β) is repeated in lines 254 and 259. In line 259 use only TGF-β as we find in Table 1.
In line 256 we find but in the references section we see “Song, Y.; Li, Z.; He, T.; Qu, M.; Jiang, L.; Li, W.; Shi, X.; Pan, J.; Zhang, L.; Wang, Y.; et al. M2 microglia-derived exosomes protect the mouse brain from ischemia-reperfusion injury via exosomal miR-124. Theranostics 2019, 9, 2910-2923, doi:10.7150/thno.30879.” For these reasons in line 256 the authors should use only Song et al.
In line 259 we find, “Additionally, Nardos G Tassew et al have found that …”. however, in the references section we see “Tassew, N.G.; Charish, J.; Shabanzadeh, A.P.; Luga, V.; Harada, H.; Farhani, N.; D'Onofrio, P.; Choi, B.; Ellabban, A.; Nickerson, 633 P.E.B.; et al. Exosomes Mediate Mobilization of Autocrine Wnt10b to Promote Axonal Regeneration in the Injured CNS. Cell 634 reports 2017, 20, 99-111, doi:10.1016/j.celrep.2017.06.009.” The citation in the middle of the text should bring the surname or the author instead of the entire name. For these reasons, please change the sentence in line 259 for ““Additionally, Tassew et al have found…”
The same problem is with “Thorsten RD et al” in line 333.
Please check along with the text if other citations should be corrected. There are some.
In line 294, MAPK is cited for the first time. Please include that it means
Mitogen-activated protein kinase.
In line 295 we see that “…ER stress-induced apoptosis and inflammation by targeting BIP [82]”. Is BIP the acronym for “binding immunoglobulin protein”? If yes, please define in the text.
In line 359 we find “…oxygen glucose deprivation (OGD)…”. OGD was not mentioned in other part of the text, so it should be removed from the text.
In line 365-366, the authors say that “In preclinical and clinical research, NSCs-based therapy has shown potential for the regenerative treatment in cerebral I/R injury”. What kind of potential? Please, elaborate.
In line 371 we see that “Pluchino et al have revealed that IFN-γ and TNF-α regulated…”. IFN-γ and TNF-α are cited here, but there are no definitions. Please include Interferon-γ and Tumor Necrosis Factor-alpha.
In line 373 we can find that “Interestingly, IFN-γ did not affect NSCs proliferation but increased the SOD2 level of…”. The authors refer to SOD for the first time here. Is the acronym for Superoxide dismutase? If yes, please, include it.
I suggest that the authors review and double-check all the citations in the middle of the text and also double-check the acronyms (abbreviations) used along with the text. Please include the definition the first time they are cited, and then, use only the acronym.
CONCLUSION and GENERAL COMMENTS
As a review article, I think it would be appropriate to include the search strategy, mesh terms used for the search and databases that were consulted to perform the review.
Although I have understood throughout the text that the use of exosomes still requires further studies in humans (since there is only one clinical trial / as a stroke treatment, in which miR-124 enriched MSC-exosomes was administrated- 420 treated in patients with acute ischemic stroke (ClinicalTrials.gov: NCT03384433)), I miss the authors including possible limitations of the use of these vesicles. Is it possible to include these limitations throughout the text or as an extra sentence in the Conclusion? Or even as a separate item before the Conclusion section?
It would also be interesting to include the limitations of this review. Are there others in the literature? If so, what are the differences for this one?
Author Response
The response to reviewer1 was added as an attachment。

Reviewer 2 Report
This topic is very interesting, but some points need to be improved:
- Lines 56-58: "In the present review, we highlight the role of exosomes in cerebral repairing processes and discuss the therapeutic impact and potential applications of exosomes for cerebral I/R injury". Are these both primary objectives? If yes, they should be both considered in the conclusion section.
- How was this review done? Is there a prisma flowchart?
- Lines 170-173: "In addition, as a key mediator of neuron-inflammation... and improved neurological deficits" Improve this concept.
- Lines 375-378: "Besides to the altered characteristics of stem cells..." What about exosomes in ischemic disease and covid-19. These papers should be considered: -- doi: 10.1166/jbn.2020.2910 -- doi: 10.3390/neurolint14020032
- Lines 288-290: "exosomes-mediated M2 polarization... which facilitate brain function recovery and improve the prognosis of stroke". Discuss more about the role of IL-4 an IL-5.
- Lines 410-418: "At present, there are still some problems that need... Lastly, studies on the treatment of cerebral I/R with exosomes are mostly 418 performed in animal models." Probably this part should be move to the discussion section. "Limits of the study" or "Future research"?
Author Response
Response to reviewer 2 was added as an attachment.

Reviewer 3 Report
This review paper provided an overview of the current literature on the possible role of cargo derived from exosomes, and exosomes-mediated intercellular communication in cerebral ischemia/reperfusion (I/R) injury. In addition, the authors aimed to share the potential therapeutic effects of stem cells-derived exosomes and underlying mechanisms in cerebral I/R injury. Overall, exosome studies reported to date have been briefly described, and excellent graphics are used to help intuitive understanding. Here are my suggestions:
1. Are Figures free from copyright issues?
2. Please quote Figures 4 and 5 in the text.
3. In Figure 4, isn't it correct that the clearance of oligodendrocyte-derived exosomes is included in astrocyte-derived exosomes? And it seems better to describe the role of microglia-derived exosomes by subtype.
4. Line 265, “in in” typing error.
5. In Table 1, explanations for abbreviations should be added.
6. In line 417, it seems that the authors intended to use "extend" rather than "extent". Is that right?
Author Response
The response to reviewer 3 was added as an attachment.

Round 2
Reviewer 2 Report
good